# Genomic Characteristics and Functional Analysis of *Brucella* sp. Strain WY7 Isolated from Antarctic Krill

**DOI:** 10.3390/microorganisms11092281

**Published:** 2023-09-11

**Authors:** Zhengqi Feng, Yuanyuan Wang, Lingbo Ma, Shanzi Huang, Lumin Wang, Jianguo He, Changjun Guo

**Affiliations:** 1State Key Laboratory for Biocontrol and Southern Marine Science and Engineering Guangdong Laboratory (Zhuhai), School of Marine Sciences, Sun Yat-sen University, 135 Xingang Road West, Guangzhou 510275, China; fengzhq6@mail2.sysu.edu.cn (Z.F.); wangyy373@mail.sysu.edu.cn (Y.W.); huangshz7@mail2.sysu.edu.cn (S.H.); lsshjg@mail.sysu.edu.cn (J.H.); 2Key Laboratory of the East China Sea and Oceanic Fishery Resources Exploitation, Ministry of Agriculture, East China Sea Fisheries Research Institute, Shanghai 116023, China; malingbo@vip.sina.com (L.M.); lmwang@eastfishery.ac.cn (L.W.); 3Guangdong Province Key Laboratory for Aquatic Economic Animals, and Guangdong Provincial Key Laboratory of Marine Resources and Coastal Engineering, Sun Yat-sen University, 135 Xingang Road West, Guangzhou 510275, China

**Keywords:** Antarctic krill, microorganism, extremophiles, *Brucella*, whole genome

## Abstract

Antarctic krill (*Euphausia superba*) is a key species of the Antarctic ecosystem whose unique ecological status and great development potential have attracted extensive attention. However, the genomic characteristics and potential biological functions of the symbiotic microorganisms of Antarctic krill remain unknown. In this study, we cultured and identified a strain of *Brucella* sp. WY7 from Antarctic krill using whole-genome sequencing and assembly, functional annotation, and comparative genomics analysis. First, based on 16S rDNA sequence alignment and phylogenetic tree analysis, we identified strain WY7 as *Brucella*. The assembled genome of strain WY7 revealed that it has two chromosomes and a plasmid, with a total genome length of 4,698,850 bp and an average G + C content of 57.18%. The DNA—DNA hybridization value and average nucleotide identity value of strain WY7 and *Brucella anthropi* ATCC^®^ 49188^TM^, a type strain isolated from human clinical specimens, were 94.8% and 99.07%, respectively, indicating that strain WY7 is closely related to *Brucella anthropi*. Genomic island prediction showed that the strain has 60 genomic islands, which may produce HigB and VapC toxins. AntiSMASH analysis results showed that strain WY7 might produce many secondary metabolites, such as terpenes, siderophores and ectoine. Moreover, the genome contains genes involved in the degradation of aromatic compounds, suggesting that strain WY7 can use aromatic compounds in its metabolism. Our work will help to understand the genomic characteristics and metabolic potential of bacterial strains isolated from Antarctic krill, thereby revealing their roles in Antarctic krill and marine ecosystems.

## 1. Introduction

The Antarctic krill is endemic in the Southern Ocean, serving as a direct energy link between the ocean’s primary producers (phytoplankton) and higher trophic levels. Krill comprises the largest population of a multicellular wild animal species on Earth [1] and plays a critical role in marine biogeochemical cycles that affect climate and ocean productivity. Antarctic krill has a unique living environment, and a large number of symbiotic microorganisms colonize its body surface, digestive tract, and other organs [2]. These symbiotic microorganisms may be closely related to important life activities such as metabolism, immunity, environmental adaptation, and host defence [3,4]. This also makes the extreme marine microorganisms in the Antarctic a natural resource pool of interest. Cui X et al. [5] and Wang Y et al. [6] isolated three phyla of culturable bacteria from Antarctic krill, including Firmicutes, Proteobacteria and Actinobacteria, and showed their affiliation with twelve genera, including *Bacillus*, *Delftia*, *Psychrobacter*, *Acinetobacter* and *Pseudoalteromonas*. An increasing number of active secondary metabolites have been found from different groups of Antarctic microorganisms, showing antibacterial, antitumour, antiviral, antioxidant, immunomodulatory and other biological activities [7,8]. However, current research on Antarctic krill mainly focuses on fishing, storage, and the processing of products [9,10], and there are few reports on the species diversity of symbiotic bacteria in Antarctic krill and their possible biological functions.

The genus *Brucella*, an α-2 subclass of Proteobacteria, is rod-shaped, aerobic, gram-negative, nonpigmented, free-living, and capable of producing acids from several carbohydrates and degrading nitrates and nitrites, taking advantage of abundant surrounding flagellar motility [11]. *Brucella* was first discovered in 1988 when Holmes et al. [11] isolated *Brucella anthropi* from human clinical specimens, mostly blood cultures. At present, *Brucella* has been found in a variety of environments, including water, soil, plants and animals. According to some published studies, *Brucella* can degrade hydrocarbons such as petroleum under extreme conditions [12]. *Brucella* strains that can efficiently degrade oil with crude oil as the sole carbon source were isolated from seabed sediments in the oil-polluted area of Xingang, Dalian [13]. A strain of *Brucella* found in the deep sea can produce new annamycin antibiotics, trienocycline H and I [14], which may have strong antitumour activity. It has also been reported that under anaerobic conditions, *Brucella* can produce hydrocarbons and act as biosurfactants (BSFs) [15]. The genus *Brucella* is known to catabolize aromatic compounds [16]. Wang X et al. [17] isolated a strain of *Brucella anthropi* from saline soil contaminated by polycyclic aromatic hydrocarbons (PAHs) that could degrade PAHs with 5% NaCl. In conclusion, previous studies have shown that *Brucella* has important application value in the degradation of aromatic hydrocarbons and the production of new antibiotics, biosurfactants and so on.

In the present study, the strain *Brucella* sp. WY7 from Antarctic krill was isolated and identified with whole-genome sequencing and assembly, functional annotation, and comparative genomics analysis.

## 2. Materials and Methods

### 2.1. Bacterial Isolation and Transmission Electron Microscopy Observations

Antarctic krill samples (58–60° W, 62–64° S) were obtained from the East China Sea Fisheries Research Institute (Shanghai, China) in November 2016 [6]. The samples were washed three times with sterile seawater to remove excess sediment and microorganisms. Three krill from the same site were ground in a mortar as a sample. The ground samples were diluted with approximately 1 mL of sterile water, collected in a 2 mL sterile centrifuge tube, and centrifuged at 3500 rpm for 5–10 min. A small amount of supernatant was obtained using the inoculation loop and applied to agar-mixed LB medium. The inoculated medium was cultured at 10 °C until colonies were visible. The colonies were randomly isolated from the medium, collected and expanded approximately three times under the same conditions. A total of 86 colonies were obtained, and the strains were stored in liquid medium containing 20% glycerol in a refrigerator at −8 °C for subsequent use. The shape of the isolated bacteria was observed by transmission electron microscopy. A small number of colonies were picked by the inoculation loop and put into a 1.5 mL centrifuge tube. Then, sterile ultrapure water was added, mixed, and centrifuged at low speed, and the cleaning was repeated 3 times. Ten microlitres of bacterial solution were adsorbed on the copper wire for observation and negatively stained with a 2% sodium phosphotungstate solution. Finally, the shape of the bacteria was observed by transmission electron microscopy.

### 2.2. DNA Extraction, PCR Amplification and Phylogenetic Tree Construction

Genomic DNA was extracted from purified and cultured strains using a bacterial DNA extraction kit (Omega, Washington, DC, USA). Using the extracted DNA of the strain as a template, the sequence of the 8–1510 bp region in bacterial 16S rDNA was amplified with universal primers 27F (5′-AGAGTTTGATCCTGGCTCA-3′) and 1492R (5′-GGTTACCTTGTTACGACTT-3′) (Tsingke, Beijing, China), and then NCBI BLAST was used for comparison. The length of the amplified fragment was approximately 1500 bp. The ClustalW program in MEGA-X was used to analyse the phylogeny [18,19,20].

### 2.3. Genome Assembly

The PacBio Sequel platform was used for single-molecule real-time (SMRT) sequencing [21]. PacBio sequencing was carried out by OE Biotech Co., Ltd. (Shanghai, China). Based on third-generation sequencing data, the genome was assembled from scratch. Falcon [22] was used for preliminary assembly, and GenomicConsensus v0.8.0 was used for correction. The SPRAI (single-pass read accuracy improver)-corrected sequence was used as auxiliary data for Circulator [23] cyclization processing, and the cyclized bacterial genome was obtained. The genome information of the bacteria was uploaded to NCBI, and accession numbers are CP049796–CP049798 for *Brucella* sp.

### 2.4. Genomic Bioinformatics Analysis

The software Prodigal (Prokaryotic Dynamic Programming Genefinding Algorithm) v2.6.3 [24] was used for coding-gene prediction. RepeatMasker v4.0.7 [25] was used to predict the repeat sequence. tRNAscan-SE v1.3.1 [26] was used to predict tRNA, RNAmmer v1.2 [27] was used to predict rRNA, and Rfam v10.0 [28] was used to predict sRNA. The preceding software parameters were default.

A genome circle map was generated using the Proksee server. The dDDH (digital DNA—DNA hybridization) values were calculated using the Type (Strain) Genome Server [29]. The JSpeciesWS online service [30] was used to calculate the ANIb (average nucleotide identity) value between genomes. Annotation was carried out using RAST (Rapid Annotation using Subsystem Technology) server 2.0 [31]. The RAST subsystem is a collection of functionally related protein families. RAST is an automated annotation server for microbial genomes built on the framework provided by the SEED system. The KEGG (Kyoto Encyclopedia of Genes and Genomes) database [32] was used for annotation. Briefly, the comparison results were compared with the KEGG database, and then KOBAS v3.0 (http://kobas.cbi.pku.edu.cn/kobas-2.1.1/kobas-3.0.3.tar.gz, accessed on 15 July 2023) [33] was used to convert the comparison results into KO and pathway annotations. CAZy (Carbohydrate Active Enzymes Database) [34] was used for annotation. The database contains families of enzymes that catalyse carbohydrate degradation, modification, and biosynthesis. Genomic islands (GIs) were predicted using IslandViewer 4 [35]. A genomic island is a kind of gene cluster that can be transferred horizontally in the genome of microorganisms. Genome islands can be predicted by differences in the genome composition of different bacterial species, mainly by SIGI-HMM, IslandPath-DIMOB and other methods. The AntiSMASH server [36] was used to predict biosynthesis-related gene clusters and perform secondary metabolic analysis.

The OrthoVenn2 server [37] was used for genomic homology cluster analysis. Venn diagrams and histograms were used to describe the comparative genomic analysis of the genome of this bacterium and its related species, showing the number and function of homologous and nonhomologous gene clusters, thus analysing the functional characteristics of this bacterium. 

## 3. Results

### 3.1. Taxonomic Identification of 16S rDNA and Morphological Characteristics

The 16S rDNA sequence alignment using NCBI BLAST showed that the sequence similarity between the isolated strain WY7 and *Brucella anthropi* ATCC^®^ 49188^TM^ was more than 99%. To further determine the phylogenetic status of the strain, a phylogenetic tree was constructed based on the 16S rDNA sequence. The results showed that the strain was closely related to *Brucella anthropi* ATCC^®^ 49188^TM^ isolated from clinical human blood [8] and *Brucella cytisi* ESC1 isolated from nodules of *Cytisus scoparius* growing in Spanish soil [38], both of which were in an independent branch (Figure 1A). Transmission electron microscopy observation showed that strain *Brucella* sp. WY7 was a bacterium with parallel sides and round ends, often appeared singly, and the size of a single cell was approximately 1.15–1.40 μm long and 0.70–0.85 μm wide (Figure 1B). These results suggested that strain WY7 might belong to the genus *Brucella*.

### 3.2. Genomic Characteristics

To obtain the genomic characteristics of strain WY7, the whole genome assembly was investigated. The assembled genome showed that strain WY7 had two chromosomes and one plasmid. As shown in Table 1, the genome length of chromosome 1 was 2,785,871 bp with 57.61% G + C content, containing 2647 genes. The genome length of chromosome 2 was 1,893,993 bp with 56.57% G + C content, containing 1718 genes. The plasmid genome length was 18,986 bp, with a G + C content of 55.17%, containing 24 genes. The total length of the genome was 4,698,850 bp, and there were three contigs with an average G + C content of 57.18%. A total of 4389 genes were identified (Figure 2A, Table 1).

To further obtain the taxonomic characteristics of strain WY7, the dDDH value and ANIb value between strain WY7 and the reference strains were calculated. The results showed that the dDDH values of strain WY7 with the type strains *Brucella anthropi* ATCC^®^ 49188^TM^ and *Brucella anthropi* NCTC^®^ 12168^TM^ were both 94.8% (Table 2), which were higher than the threshold of species definition by 70%. The ANIb values of strain WY7 and its close relatives *Brucella anthropi* ATCC^®^ 49188^TM^ and *Brucella anthropi* NCTC^®^ 12168^TM^ were as high as 99.07% (Table 3), higher than the threshold of 95% for defining species. These results indicated that strain WY7 is closely related to *Brucella anthropi*.

### 3.3. RAST Quick Genome Annotation

The functional distribution of strain WY7 was preliminarily obtained by genome annotation with RAST. According to the SEED system of RAST, genes were assigned to subsystems, which could be divided into 27 categories. The genome annotation information showed that RAST divided them into 339 subsystems; only 28% (1256 genes) were annotated in the subsystem, and the other 72% of the genome was not assigned to the RAST subsystem. Among them, Amino Acids and Derivatives (359 genes), Carbohydrates (228 genes), Protein Metabolism (212 genes), Cofactors, Vitamins, Prosthetic Groups, Pigments (153 genes), Respiration (108 genes) and other functions accounted for the majority (Figure 2B). The above results showed that strain WY7 was rich in amino acids and their derivatives, carbohydrates and protein metabolism.

Notably, genes related to Cell Division and Cell Cycle were not found in strain WY7, but a gene related to Dormancy and Sporulation was found. The genome also lacked genes related to Photosynthesis, suggesting that the bacterium was not a photosynthetic bacterium. 

### 3.4. KEGG Database Annotation

The genome function was annotated using the KEGG database, and the approximate functional distribution of strain WY7 was obtained. Gene functions were divided into five categories by the KEGG database: Cellular Process, Environmental Information Processing, Genetic Information Processing, Metabolism, and Organismal Systems, and then each category was further subdivided into subcategories (Figure 3A, Table 4). Among them, 1419 genes were annotated on chromosome 1, 861 genes were annotated on chromosome 2, and only one gene was annotated on the plasmid, whose function was gene replication and repair.

On chromosome 1, amino acid metabolism (159 genes), global and overview maps (156 genes), metabolism of cofactors and vitamins (132 genes), carbohydrate metabolism (126 genes), membrane transport (125 genes), and energy metabolism (109 genes) were the dominant functional categories. On chromosome 2, membrane transport (176 genes), cellular community—prokaryotes (106 genes), carbohydrate metabolism (104 genes) and other functional categories dominated. A total of 40 genes in WY7 were associated with organismal systems, mainly involving the endocrine system, aging, environmental adaptation, nervous system, circulatory system, and digestive system. The results showed that strain WY7 has strong carbohydrate metabolism and membrane transport capacity and might also have an impact on the organizational systems of its host, thus providing great potential for environmental adaptation and the degradation, transformation, and utilization of complex carbohydrates.

In-depth analysis of the metabolic pathways of strain WY7 revealed genes related to aromatic hydrocarbon degradation pathways, such as naphthalene 1,2-dioxygenase ferredoxin reductase component (gene 0212 and gene 2319), protocatechuate 3,4-dioxygenase (gene 0594 and gene 0595), and salicylate hydroxylase (gene 0845). In addition, we mapped the relevant pathways of aromatic hydrocarbons (e.g., salicylate, 2,4-dinitrotoluene, nitrobenzene, naphthalene, and 1-methylnaphthalene) that suggest the isolate may be involved in degradation (Figure 3B). The above results indicated that strain WY7 has potential for application in the process of aromatic hydrocarbon metabolism.

### 3.5. CAZy Database Annotation

To explore the characteristics of the carbohydrate-active enzyme family of WY7, the predicted genes were annotated using the CAZy database (Figure 4A). Twenty-one genes encoding glycosyl transferases (GTs), 18 genes encoding glycoside hydrolases (GHs), 11 genes encoding carbohydrate esterases (CEs) and one gene encoding carbohydrate binding modules (CBMs) were identified on chromosome 1. Fourteen genes encoding GT, seven genes encoding GH and five genes encoding CE were identified on chromosome 2. Only one gene encoding GH was identified in the plasmid. These observations indicated that glycosyl transferases and glycoside hydrolases accounted for the majority in strain WY7, providing a basis for the formation, transfer and further metabolism of monosaccharides, polysaccharides, and glycosides.

### 3.6. Genomic Islands Prediction

To investigate more potential special functions of *Brucella* sp. WY7, 25 genomic islands on chromosome 1, 35 genomic islands on chromosome 2, and their location in the genome of the strain were predicted by IslandViewer4 (Figure 4B). The 25 genomic islands of chromosome 1 were composed of 484 genes, with gene distributions ranging from 295,164 to 2,494,506 bp. Among them, 221 genes expressed hypothetical nonfunctional proteins, eight genes expressed mobile element proteins, five genes located on the same genomic island expressed anti-restriction proteins, and two genes located on the same genomic island expressed VapC toxin proteins. The 35 genomic islands of chromosome 2 were composed of 464 genes, which ranged from 216,887 to 1,878,563 bp. Seven genes were found to express mobile element proteins and two HigB toxin genes were found to be located on the same genomic island. These results suggested that the strain had gene clusters transferred horizontally and that some gene clusters could express toxin proteins involved in the regulation of bacterial growth under specific environments.

### 3.7. Secondary Metabolites

To gain further insight into the potential functions of WY7, secondary metabolites were predicted by antiSMASH. Gene screening for secondary metabolites on chromosome 1 revealed four distinct gene clusters (Figure 5). The secondary metabolites were batalactone, arylpolyene, terpene and acyl amino acids. Gene Clusters 1, 2 and 3 could not identify the known homologous gene cluster, and gene Cluster 4 (nucleotide 1,939,685–2,000,431) had 25% similarity with the gene cluster NRP producing ambactin.

Gene screening of secondary metabolites on chromosome 2 revealed three different gene clusters (Figure 5), and the secondary metabolites were siderophore, NAGGN and ectoine. Neither Cluster 2 nor Cluster 3 identified known homologous clusters, and Cluster 1 (nucleotide 1,159,292–1,173,812) was 85% similar to a gene cluster producing Ochrobactin, which is a photoreactive amphiphilic siderophore that has been found in *Brucella* sp. SP18 and *Vibrio* sp. DS40M5 [39]. Notably, the low similarity of predicted gene clusters may represent the generation of new metabolites.

### 3.8. Pan-Genomic Comparative Analysis

To acquire comparative genomic characteristics of WY7, homogenous cluster analysis of the genome was performed using OrthoVenn2. This Venn diagram and histogram describe the comparative genomics of *Brucella* sp. WY7 and *Brucella anthropi* ATCC^®^ 49188^TM^, *Brucella anthropi* PBO, *Brucella anthropi* NCTC^®^ 12168^TM^ and *Brucella melitensis* bv. 1 str. 16M (Figure 6). The genome in the overlapping centre was composed of 1101 homologous clusters. Most of the annotation functions of homologous clusters were involved in biological processes, such as cellular processes, primary metabolic processes, cellular aromatic compound metabolic processes, organic acid metabolic processes and macromolecule metabolic processes. Strain *Brucella* sp. WY7 was found to have only two nonshared gene clusters, involving processes that activate or increase the frequency, rate or extent of cellular DNA-templated transcription. The existence of nonshared gene clusters indicated that the biological process of this strain might have unique advantages over other related strains.

The specific primers (F-5′-GTCAGTCGGCGGCTTATTC-3′ and R-5′-ATATCGTCTCCTTGGCAATGTC-3′) of gene 1983 of chromosome 1 in the WY7 nonshared gene clusters were designed for PCR detection of *Brucella* sp. WY7 and its relatives, *Brucella anthropi* ATCC^®^ 49188^TM^ (Appendix A). The results show that only WY7 has the correct band, which means that *Brucella anthropi* ATCC^®^ 49188^TM^ does not contain this gene, and *Brucella* sp. WY7 we found in Antarctic krill is not completely the same as its relatives.

## 4. Discussion

The Antarctic Ocean has the largest proportion of unknown genes to date of all sea areas. Most reports on psychrophilic bacteria in marine ecosystems are limited to sea ice or deep-sea sediments, while the study of Antarctic heterotrophic isolated bacteria shows that the growth rate of bacteria is quite high in the temperature range of 0–20 °C [40]. The study found that the proportion of functional housekeeping genes related to cell growth and metabolism is as high as 85–90%. This advantage of housekeeping genes shows that a few key genes that determine which species are dominant are of disproportionate importance. Among these dominant groups, some genes related to secondary metabolism have been found, which has potential importance for biological exploration [40]. However, we know little about the factors that control the distribution and dominance of key microbial species, but finding these key genes may be an achievable goal. In this study, we isolated and identified the strain *Brucella* sp. WY7, belonging to the genus not previously discovered in Antarctica, from Antarctic krill and explored its genomic characteristics and potential functions.

Defining a new species involves two successive steps, which are 16S rDNA analysis and calculation of genomic parameters, and the minimum standards (<98.7%) proposed by Chun et al. for prokaryotic classification using genomic data [41]. The threshold values of dDDH and ANIb for species division were 70% and 95%, respectively. If the value obtained by comparing the genome of this bacterium with that of a closely related type strain is lower than the threshold value of species division, it may be a new species [30]. The ANIb and dDDH values of *Brucella* sp. WY7 and *Brucella anthropi* ATCC^®^ 49188^TM^ were 99.07% and 94.8%, respectively, both of which are significantly higher than the threshold of species division. This indicated that this strain may be the same species as *Brucella anthropi* ATCC^®^ 49188^TM^ and may be *Brucella anthropi*, which has not been found in Antarctica before. Specific primers were designed, and the grinding solution of Antarctic krill and the genomic DNA of *Brucella* sp.WY7 were used as templates for PCR amplification to eliminate the effects of experimental operation and environmental pollution. The results showed that the amplified product fragment was consistent with the size of the target gene, which verified the authenticity of the isolated strain (Appendix A). 

Research has shown that aromatic compounds are among the most prevalent and persistent environmental pollutants [42]. The genus *Brucella* is known to catabolize aromatic compounds [16]. In this study, genes and pathways associated with aromatic compound metabolism were found, such as the naphthalene 1,2-dioxygenase ferredoxin reductase component gene, protocatechuate 3,4-dioxygenase gene, and salicylate hydroxylase gene. Notably, salicylate hydroxylase (E1.14.13.1) is the first enzyme in the downstream pathway of naphthalene degradation [43], and protocatechuate 3,4-dioxygenase (pcaG and pcaH) is a key enzyme that catalyses the ring-opening step of aromatic compounds in bacteria [44]. The above results indicate that strain WY7 has potential for application in the process of aromatic hydrocarbon metabolism. The potential secondary metabolites of *Brucella* sp. WY7 were analysed, and the gene clusters of betalactone, arylpolyene, terpene, acyl amino acids, siderophores, NAGGN and ectoine were found. The similarity between the gene cluster producing siderophore and a gene cluster producing ochrobactin was 85%. Siderophores are important bioactive substances that can transport extracellular iron ions for bacterial use [45]. Terpenoids have often been extracted from marine organisms in recent years. They have important physiological activities and are important sources for developing new drugs and studying natural products [46]. Ectoine is an osmotic regulator that helps stabilize the structure of microorganisms and enables them to adapt to extreme environments [47]. Horizontal transfer of mobile elements in microbial communities is an important mechanism by which bacterial genomes evolve and adapt to specific environmental stresses [48]. It has been reported that HigB toxin and VapC toxin act as endribosomal enzymes to specifically cleave mRNA and tRNA in bacteria, thereby blocking transcription and translation processes and inhibiting bacterial growth. These toxins often coexist with antitoxins to regulate bacterial growth and adaptation to extreme environments [49,50]. The prediction of genome islands revealed that the bacterium had 15 mobile element protein genes, which might express anti-restriction proteins and produce HigB toxin and VapC toxin. It is speculated that the above genes help bacteria adapt to extreme environments.

By identifying homologous gene clusters from ancestors, comparative analyses can help clarify the relationships between different species and the evolution and adaptation of genomes [51]. The experimental strain *Brucella* sp. WY7 and four related strains had 4815 clusters and 1153 nonshared clusters. The distribution of core partial functions indicates that most homologous gene families encode basic bacterial metabolism, such as hydrolase activity, transferase activity, protein processing, folding and secretion, and DNA metabolism. Strain *Brucella* sp. WY7 had only two nonshared clusters, with functions in the positive regulation of transcription, indicating that its biological processes have some different characteristics compared with those of related strains; those characteristics, it is speculated, are related to its living environment. However, this needs to be further explored.

## 5. Conclusions

In this study, the whole genome of *Brucella* sp. WY7 was assembled, and the genomic characteristics were described by genomic analysis. This strain was identified within the genus *Brucella* in the class Alphaproteobacteria, which is closely related to *Brucella anthropi*. Genes involved in the degradation of aromatic compounds were identified, indicating the potential metabolism of an aromatic compound of *Brucella* sp. WY7. Through the prediction of secondary metabolites, genomic island function and comparative genomic analysis with related bacteria, the potential biological function was predicted. Our work will help to enrich the microbial resource database and provide more material sources for modern genetic engineering technology.

## Figures and Tables

**Figure 1 microorganisms-11-02281-f001:**
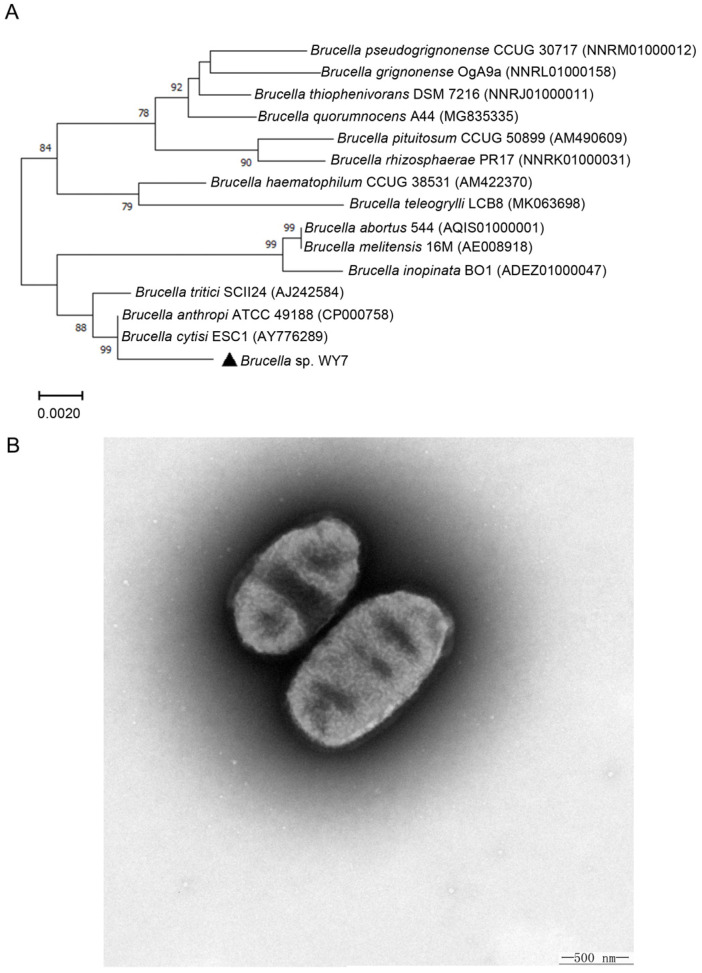
(**A**) N-J phylogenetic tree based on the 16S rDNA sequence. Numbers at nodes indicate the levels of bootstrap support (>70%) based on 1000 resamplings. Bar, 0.01 substitutions per nucleotide position. (**B**) Transmission electron micrograph showing cells of strain *Brucella* sp. WY7. The magnification is 6000×. Bar, 500 nm.

**Figure 2 microorganisms-11-02281-f002:**
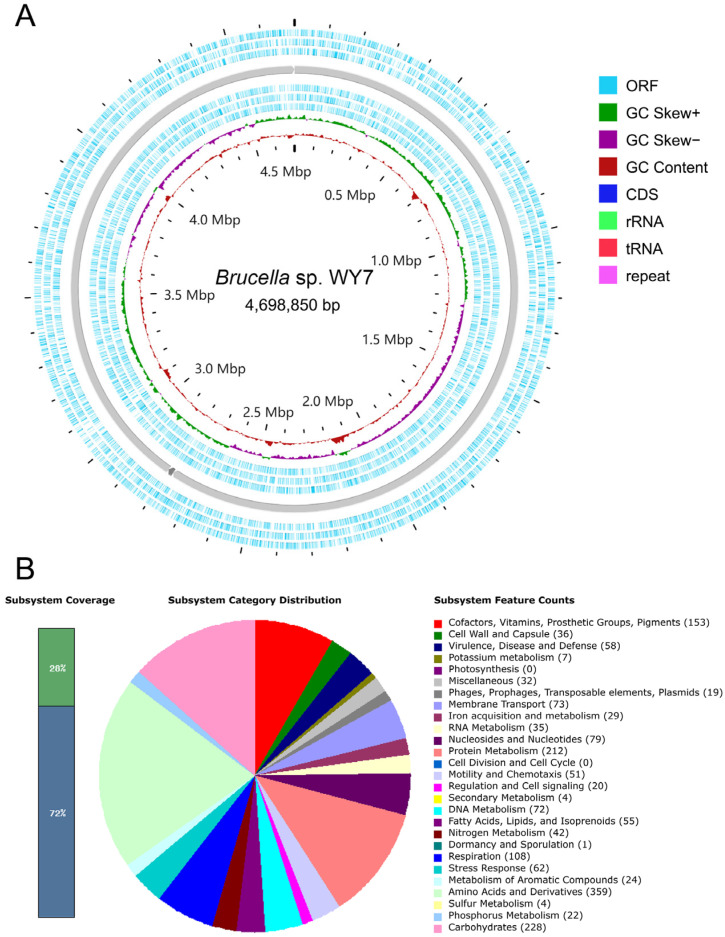
(**A**) Genome circle map of strain *Brucella* sp. WY7: GC content (oxblood red), GC skew curves (+/−, dark green/purple), open reading frames (ORFs, light blue), coding sequences (CDSs, dark blue), rRNAs (light green), tRNAs (vermilion), and repeats (pink). (**B**) RAST subsystems category distribution of annotated genes of the strain *Brucella* sp. WY7.

**Figure 3 microorganisms-11-02281-f003:**
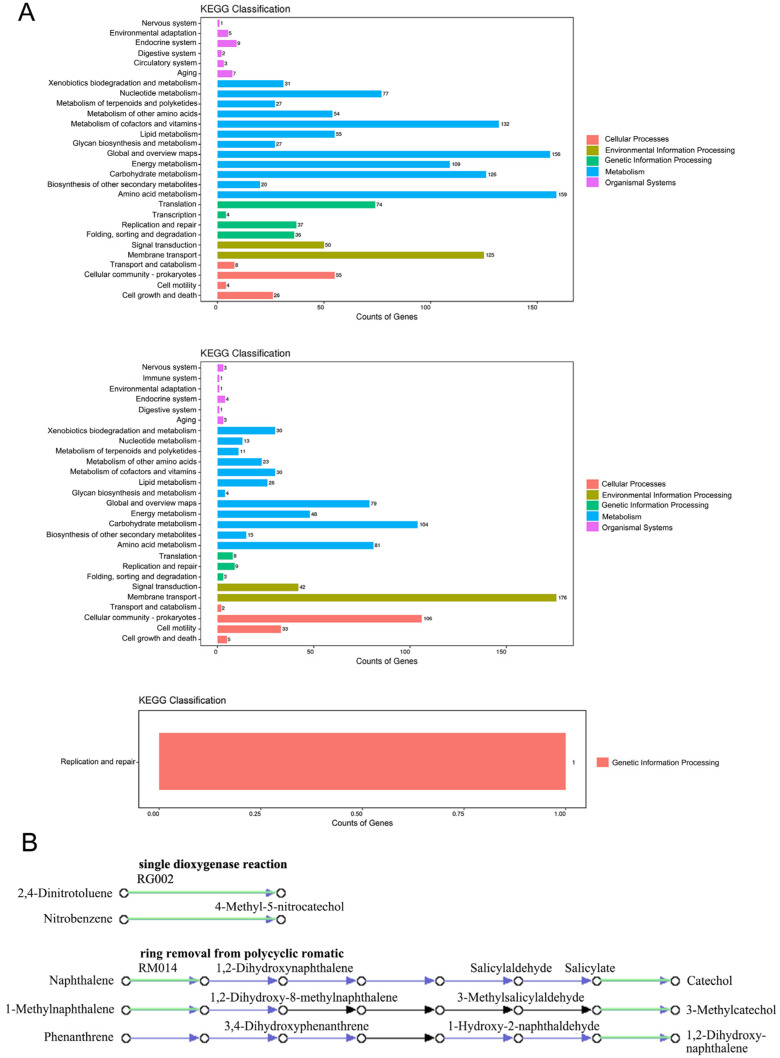
(**A**) KEGG annotation statistics chart of *Brucella* sp. WY7. The figure shows chromosome 1, chromosome 2 and the plasmid of *Brucella* sp. WY7. The horizontal axis is the number of genes, and the vertical axis represents the name of the level 2 pathway. (**B**) Aromatic hydrocarbon degradation pathways in WY7.

**Figure 4 microorganisms-11-02281-f004:**
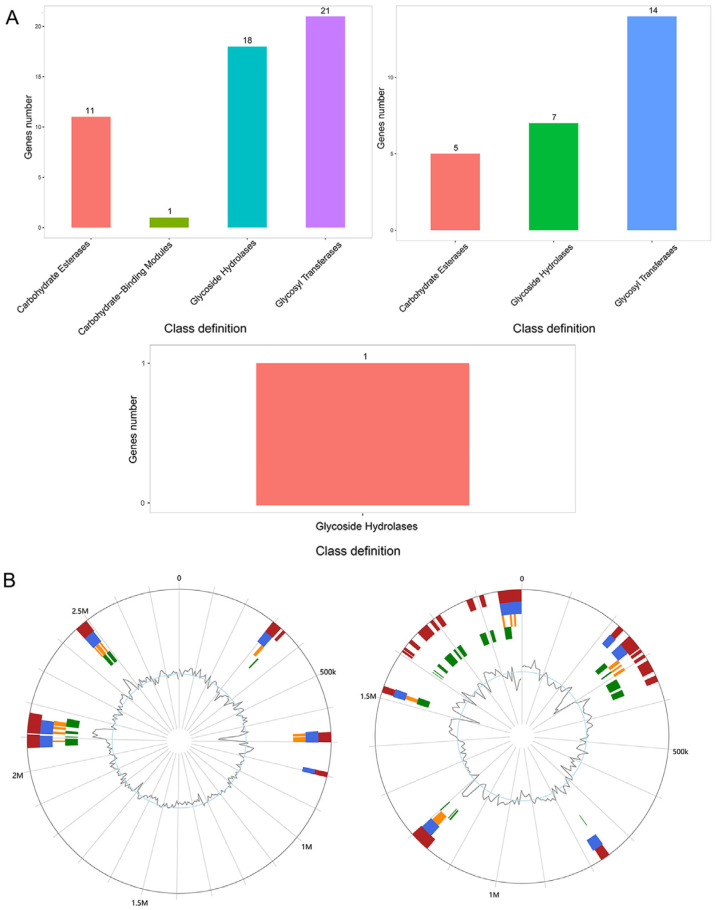
(**A**) CAZy annotation classification distribution map. The figure shows chromosome 1, chromosome 2 and plasmid of *Brucella* sp. WY7. The horizontal axis is the CAZy category, and the vertical axis is the number of genes annotated by the corresponding category. (**B**) The predicted genomic islands (GIs) of the strain *Brucella* sp. WY7. Figure shows chromosome 1 and chromosome 2. Red shows the prediction by the integrated approach; blue represents the results from IslandPath-DIMOB; orange displays genomic islands predicted via SIGI-HMM; green shows the results from IslandPick.

**Figure 5 microorganisms-11-02281-f005:**
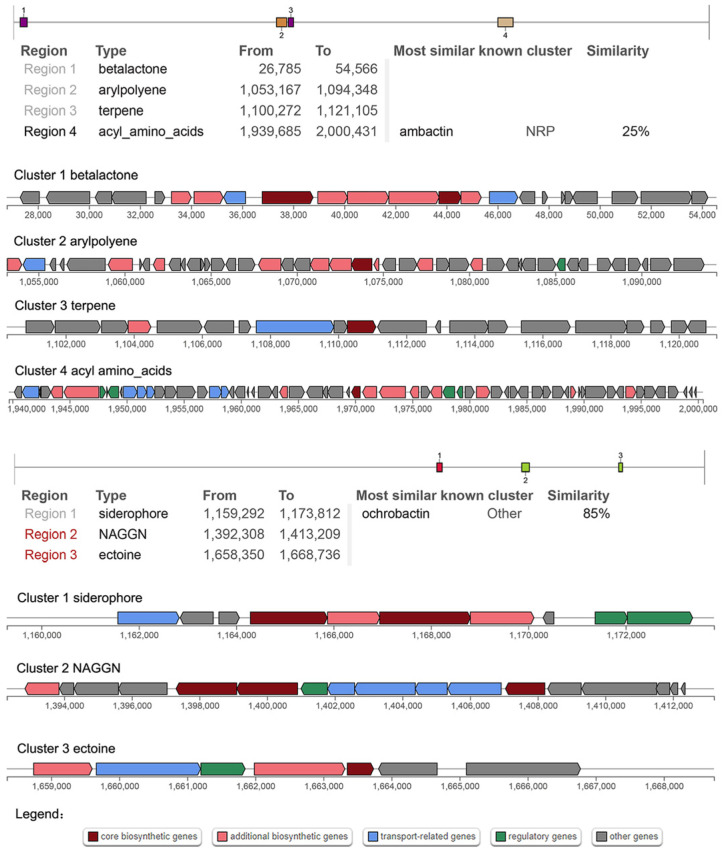
AntiSMASH predicted biosynthesis-related gene clusters and secondary metabolism. The figure shows chromosome 1 and chromosome 2 of *Brucella* sp. WY7. Brown represents the core biosynthetic genes; pink represents the additional biosynthetic genes; blue represents the transport-related genes; green represents the regulatory genes; and grey represents other genes.

**Figure 6 microorganisms-11-02281-f006:**
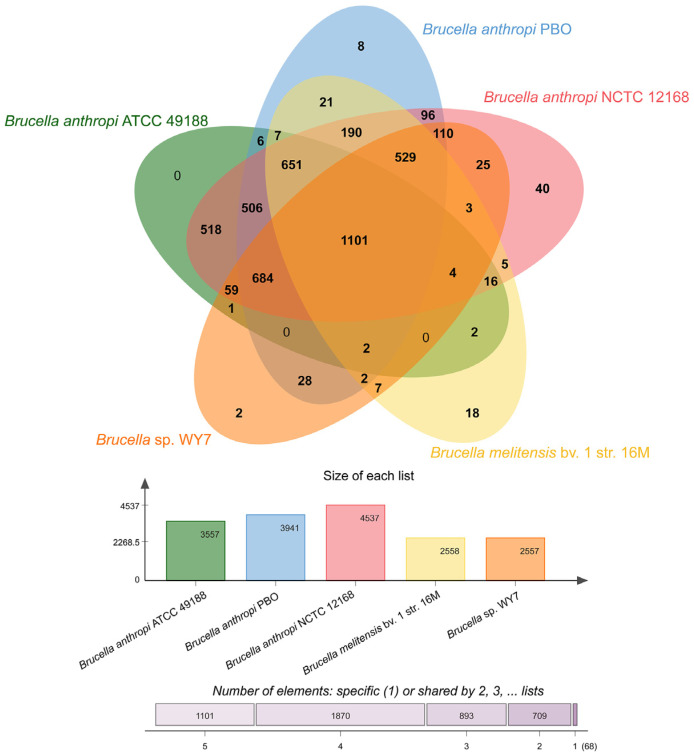
The Venn diagram and the bar graph depict the comparative genomics among the genomes of *Brucella anthropi* ATCC^®^ 49188^TM^ (green), *Brucella anthropi* PBO (blue), *Brucella anthropi* NCTC^®^ 12168^TM^ (pink), *Brucella melitensis* bv. 1 str. 16M (yellow), and *Brucella* sp. WY7 (orange), showing shared and unshared orthologous gene clusters.

**Table 1 microorganisms-11-02281-t001:** Genome data of *Brucella* sp. WY7.

Attribute	Chr.1	Chr.2	Plas.	Total
Genome size (Mb)	2.79	1.89	0.019	4.70
DNA coding region (Mb)	2.44	1.70	0.017	4.16
DNA G + C (%)	57.61	56.57	55.17	57.18
DNA scaffolds	1	1	1	3
Total genes	2647	1718	24	4389
RNA gene	47	21	0	68
Genes assigned to COGs	2230	1432	7	3669
Repeats	150	79	0	229

Chr. stands for chromosome, Plas. stands for plasmid.

**Table 2 microorganisms-11-02281-t002:** The dDDH values are provided along with their confidence intervals (C.I.).

Subject Strain	Query Strain	dDDH (d_4_, in%)	C.I. (d_4_, in%)	The Origin of Subject Strain
*Brucella anthropi* ATCC 49188 (JAAVLS010000001.1)	WY7	94.8	[93.2–96.1]	Type strain from human clinical specimens, blood cultures
*Brucella anthropi* NCTC 12168 (UGSA01000001.1)	WY7	94.8	[93.2–96.1]	Type strain from human clinical specimens, blood cultures
*Brucella lupini* LUP21 (GCA 002252535.1)	WY7	82.5	[79.7–85.1]	Plant root nodule samples
*Brucella anthropi* PBO (GCA 015326295.1)	WY7	78.8	[75.9–81.5]	Plastic debris from sea coast, Qingdao, China
*Brucella melitensis* 16M (GCA 000007125.1)	WY7	25.0	[22.7–27.5]	An infected goat

**Table 3 microorganisms-11-02281-t003:** Calculation of ANIb values between available genomes of WY7 and type strains.

Species	WY7	*Brucella anthropi* NCTC 12168	*Brucella anthropi* ATCC 49188	*Brucella lupini* LUP21	*Brucella anthropi* PBO	*Brucella melitensis* 16M
WY7	100	99.07	99.07	97.49	97.21	80.55
*Brucella anthropi* NCTC 12168	98.91	100	99.78	97.47	96.73	80.40
*Brucella anthropi* ATCC 49188	98.70	99.77	100	97.26	96.55	80.15
*Brucella lupini* LUP21	96.68	96.77	96.60	100	96.63	79.91
*Brucella anthropi* PBO	97.14	96.89	96.73	97.43	100	80.43
*Brucella melitensis* 16M	80.99	81.00	80.98	80.89	81.07	100

**Table 4 microorganisms-11-02281-t004:** Annotation function classification of *Brucella* sp. WY7 of the genomic KEGG database.

Classification	Number of Genes		
	Chr.1	Chr.2	Plas.
**Cellular Processes**			
Cell growth and death	26	5	0
Cell motility	4	33	0
Cellular community—prokaryotes	55	106	0
Transport and catabolism	8	2	0
**Environmental Information Processing**			
Membrane transport	125	176	0
Signal transduction	50	42	0
**Genetic Information Processing**			
Folding, sorting and degradation	36	3	0
Replication and repair	37	9	1
Transcription	4	0	0
Translation	74	8	0
**Metabolism**			
Amino acid metabolism	159	81	0
Biosynthesis of other secondary metabolites	20	15	0
Carbohydrate metabolism	126	104	0
Energy metabolism	109	48	0
Global and overview maps	156	79	0
Glycan biosynthesis and metabolism	27	4	0
Lipid metabolism	55	26	0
Metabolism of cofactors and vitamins	132	30	0
Metabolism of other amino acids	54	23	0
Metabolism of terpenoids and polyketides	27	11	0
Nucleotide metabolism	77	13	0
Xenobiotics biodegradation and metabolism	31	30	0
**Organismal Systems**			
Aging	7	3	0
Circulatory system	3	0	0
Digestive system	2	1	0
Endocrine system	9	4	0
Environmental adaptation	5	1	0
Immune system	0	1	0
Nervous system	1	3	0
**Total**	1419	861	1

## Data Availability

The authors confirm that the data supporting the findings of this study are available within the article and its Appendix A.

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
