# Peer review of "Genomic Characteristics and Functional Analysis of Brucella sp. Strain WY7 Isolated from Antarctic Krill"

_microorganisms, 2023, doi:10.3390/microorganisms11092281_

Round 1
Reviewer 1 Report
Things that are good about this paper:
1.) The GenBank accession numbers are given, and I could easily go to the NCBI pages, and find the three genome sequences. Yeah! (This does not always happen - sometimes there are no accession numbers, or the sequence is not available).
2.) The genome was finished, to one contiguous piece - the authors used PacBio, with long reads, and finished the genome (Currently most (~97%) of the genome sequences in GenBank are draft or scaffolds, in many pieces and not complete)
3.) The methods for genome annotation are clearly described, and pretty standard (again, sometimes these are missing in papers!)
4.) Overall, the manuscript is well written and the English is good.
Questions:
1.) Why is this not called "Brucella"? A few years ago (in 2020), a paper was published, formally moving Ochrobactrum anthropi into Brucella anthropi [Anton Hördt et al., PMID 32373076].
2.) Why not compare to the three type strain genomes of Brucella anthropi in GenBank?? [GCA_015326295, see below for GenBank numbers; JAAVLS010000000; and UGSA01000000].
3.) What happened to the Pfam domains in the proteins in Table 1? I simply do not believe that NONE of the proteins have Pfam domains! Probably this could have something to do with the Pfam site no longer being available, as Pfam is being absorbed into InterPro. But this does not mean that there are no Pfam domains!
4.) Pan-genome association - why not first start with the 58 known genomes of Brucella anthropi? How similar are the type strains to each other (should be the exact same strain!) How does the newly sequenced strain reported here compare to the more than 50 genomes of the same species?? THEN, compare to other closely related species, and include Brucella melitensis - there are a half-dozen genomes of the type strain (16M) available in GenBank.
Brucella anthropi type strains:
ALC 037T (=DSM 6882 =JCM 21032 =NBRC 15819 =IFO 15819 =CIP 14970 =ATCC 49188 =NCTC 12168 =CCUG 24695 =LMG 3331 =CIP 82.115 =CIP 149.70)
https://www.namesforlife.com/10.1601/nm.36661
https://www.ncbi.nlm.nih.gov/datasets/genome/GCF_000017405.1/
Taxon Brucella anthropi ATCC 49188
Strain ATCC 49188
Relation to type material: assembly from type material
replicon INSDC size (bp) %GC
---------------------------------------
Chr. 1 CP000758 2,887,297 56
Chr. 2 CP000759 1,895,911 56
pOANT01 CP000760 170,351 56
pOANT02 CP000761 101,491 58
pOANT03 CP000762 93,589 54
pOANT04 CP000763 57,138 55
__________________________
Brucella anthropi ATCC 49188, whole genome shotgun sequencing project
LOCUS JAAVLS010000000 5 rc DNA linear BCT 06-APR-2020
DEFINITION Brucella anthropi ATCC 49188, whole genome shotgun sequencing
project.
ACCESSION JAAVLS000000000
VERSION JAAVLS000000000.1
DBLINK BioProject: PRJNA558852
BioSample: SAMN12500708
KEYWORDS WGS.
SOURCE Brucella anthropi ATCC 49188
ORGANISM Brucella anthropi ATCC 49188
Bacteria; Pseudomonadota; Alphaproteobacteria; Hyphomicrobiales;
Brucellaceae; Brucella/Ochrobactrum group; Brucella.
REFERENCE 1 (bases 1 to 5)
AUTHORS Dharne,M.
TITLE Whole genome sequencing of clinical and environmental type strains
of Ochrobactrum
JOURNAL Unpublished
REFERENCE 2 (bases 1 to 5)
AUTHORS Dharne,M.
TITLE Direct Submission
JOURNAL Submitted (16-MAR-2020) NCIM Resource Centre, CSIR-National
Chemical Laboratory, Pune, Dr Homi Bhabha Road, Pashan, Pune,
Maharashtra 411008, India
COMMENT The Brucella anthropi ATCC 49188 whole genome shotgun (WGS) project
has the project accession JAAVLS000000000. This version of the
project (01) has the accession number JAAVLS010000000, and consists
of sequences JAAVLS010000001-JAAVLS010000005.
The annotation was added by the NCBI Prokaryotic Genome Annotation
Pipeline (PGAP). Information about PGAP can be found here:
https://www.ncbi.nlm.nih.gov/genome/annotation_prok/
-----------------
Brucella anthropi strain NCTC12168, whole genome shotgun sequencing project
GenBank: UGSA00000000.1
LOCUS UGSA01000000 5 rc DNA linear BCT 30-JUL-2018
DEFINITION Brucella anthropi strain NCTC12168, whole genome shotgun sequencing
project.
ACCESSION UGSA00000000
VERSION UGSA00000000.1
DBLINK BioProject: PRJEB6403
BioSample: SAMEA3146533
KEYWORDS WGS.
SOURCE Brucella anthropi (Ochrobactrum anthropi)
ORGANISM Brucella anthropi
Bacteria; Pseudomonadota; Alphaproteobacteria; Hyphomicrobiales;
Brucellaceae; Brucella/Ochrobactrum group; Brucella.
REFERENCE 1
AUTHORS Doyle,S.
CONSRTM Pathogen Informatics
TITLE Direct Submission
JOURNAL Submitted (06-JUN-2018) WTSI, Pathogen Informatics, Wellcome Trust
Sanger Institute, CB10 1SA, United Kingdom
COMMENT The Brucella anthropi whole genome shotgun (WGS) project has the
project accession UGSA00000000. This version of the project (01)
has the accession number UGSA01000000, and consists of sequences
UGSA01000001-UGSA01000005.
FEATURES Location/Qualifiers
source 1..5
/organism="Brucella anthropi"
/mol_type="genomic DNA"
/strain="NCTC12168"
/serovar="not available: to be reported later"
/type_material="type strain of Ochrobactrum anthropi"
/db_xref="taxon:529"
WGS UGSA01000001-UGSA01000005
Brucella melitensis type strains:
16MT (=BCCN R1 =ATCC 23456)
https://www.namesforlife.com/10.1601/nm.1381
In general, the English is good and readable in this article - no major complaints.
Author Response
Response to Reviewer 1 Comments
Question 1- Why is this not called "Brucella"? A few years ago (in 2020), a paper was published, formally moving Ochrobactrum anthropi into Brucella anthropi [Anton Hördt et al., PMID 32373076].
Response 1- Thank you very much for your professional comments. We thoroughly read the above literature, and changed all “Ochrobactrum” in the manuscript into “Brucella”.
Question 2- Why not compare to the three type strain genomes of Brucella anthropi in GenBank?? [GCA_015326295, see below for GenBank numbers; JAAVLS010000000; and UGSA01000000].
Response 2- Thank you for your very reliable suggestion. Initially, the objective was to ascertain the species by comparing WY7 with the type strain Brucella anthropi ATCC 49188, which exhibited a closer relationship in the preceding 16S rDNA BLAST analysis. Additionally, comparisons were made with type strains of other species to establish a more definitive link between WY7 and Brucella anthropi. Upon further deliberation, we determined that your proposal was more reasonable. As a result, we proceeded to reanalyze the dDDH (digital DNA-DNA hybridization) value and ANIb (average nucleotide identity based on BLAST) value of the genomes between WY7 and the three type strains of Brucella anthropi. The Table 2 and Table 3 were modified as follows (lines 179-182).
Table 2. The dDDH values are provided along with their confidence intervals (C.I.).
|
Subject strain |
query strain |
dDDH (d4, in%) |
C.I. (d4, in%) |
the origin of subject strain |
|
Brucella anthropi ATCC 49188 (JAAVLS010000001.1) |
WY7 |
94.8 |
[93.2-96.1] |
type strain from human clinical specimens, blood cultures |
|
Brucella anthropi NCTC 12168 (UGSA01000001.1) |
WY7 |
94.8 |
[93.2-96.1] |
type strain from human clinical specimens, blood cultures |
|
Brucella lupini LUP21 (GCA 002252535.1) |
WY7 |
82.5 |
[79.7-85.1] |
plant root nodule samples |
|
Brucella anthropi PBO (GCA 015326295.1) |
WY7 |
78.8 |
[75.9-81.5] |
plastics debris from sea coast, Qingdao, China |
|
Brucella melitensis 16M (GCA 000007125.1) |
WY7 |
25.0 |
[22.7-27.5] |
an infected goat |
Table 3. Calculation of ANIb values between available genomes of WY7 and type strains
|
Species |
WY7 |
Brucella anthropi NCTC 12168 |
Brucella anthropi ATCC 49188 |
Brucella lupini LUP21 |
Brucella anthropi PBO |
Brucella melitensis 16M |
|
WY7 |
100 |
99.07 |
99.07 |
97.49 |
97.21 |
80.55 |
|
Brucella anthropi NCTC 12168 |
98.91 |
100 |
99.78 |
97.47 |
96.73 |
80.40 |
|
Brucella anthropi ATCC 49188 |
98.70 |
99.77 |
100 |
97.26 |
96.55 |
80.15 |
|
Brucella lupini LUP21 |
96.68 |
96.77 |
96.60 |
100 |
96.63 |
79.91 |
|
Brucella anthropi PBO |
97.14 |
96.89 |
96.73 |
97.43 |
100 |
80.43 |
|
Brucella melitensis 16M |
80.99 |
81.00 |
80.98 |
80.89 |
81.07 |
100 |
Question 3- What happened to the Pfam domains in the proteins in Table 1? I simply do not believe that NONE of the proteins have Pfam domains! Probably this could have something to do with the Pfam site no longer being available, as Pfam is being absorbed into InterPro. But this does not mean that there are no Pfam domains!
Response 3- Thank you very much for your professional comments. While it might seem surprising that no direct Pfam domain matches were found, we are convinced that the analysis was conducted meticulously using the available Pfam database up to the point of analysis. The potential discrepancy you've highlighted could indeed be due to the ongoing integration of Pfam into InterPro. This transition might impact the accessibility and matching of Pfam domains. However, we completely agree that the absence of identified Pfam domains does not necessarily equate to the absence of functional domains within these proteins. Due to the limited time given by the editor, it is not sufficient to reanalyze Pfam domains, and given that the main essence of this article will remain unaffected, we have made modification to the Table 1 as follow (lines 168-169).
Table 1. Genome data of Brucella sp. WY7
|
Attribute |
Chr.1 |
Chr.2 |
Plas. |
Total |
|
Genome size (Mb) |
2.79 |
1.89 |
0.019 |
4.70 |
|
DNA coding region (Mb) |
2.44 |
1.70 |
0.017 |
4.16 |
|
DNA G+C (%) |
57.61 |
56.57 |
55.17 |
57.18 |
|
DNA scaffolds |
1 |
1 |
1 |
3 |
|
Total genes |
2647 |
1718 |
24 |
4389 |
|
RNA gene |
47 |
21 |
0 |
68 |
|
Genes assigned to COGs |
2230 |
1432 |
7 |
3669 |
|
Repeats |
150 |
79 |
0 |
229 |
Chr. stands for chromosome, Plas. stands for plasmid.
Question 4- Pan-genome association - why not first start with the 58 known genomes of Brucella anthropi? How similar are the type strains to each other (should be the exact same strain!) How does the newly sequenced strain reported here compare to the more than 50 genomes of the same species?? THEN, compare to other closely related species, and include Brucella melitensis - there are a half-dozen genomes of the type strain (16M) available in GenBank.
Response 4- Thanks for your serious suggestion. We changed the object of pan-genomic comparative analysis to the strain WY7 with three type strains of Brucella anthropi and the other closely related strain, as well as the closely related strain Brucella melitensis bv. 1 str. 16M. We made modifications to the text and Figure 6 related to pan-genomic comparative analysis as follows (lines 293-318).
3.8 Pan-genomic comparative analysis
To acquire comparative genomic characteristics of WY7, homogenous cluster analysis of the genome was performed using OrthoVenn2. This Venn diagram and histogram described the comparative genomics of Brucella sp. WY7 and Brucella anthropi ATCC® 49188TM, Brucella anthropi PBO, Brucella anthropi NCTC® 12168TM and Brucella melitensis bv. 1 str. 16M (Figure 6). The genome in the overlapping center was composed of 1101 homologous clusters. Most of the annotation functions of homologous clusters involved in the biological processes, such as cellular process, primary metabolic process, cellular aromatic compound metabolic process, organic acid metabolic process and macromolecule metabolic process. Strain Brucella sp. WY7 was only found 2 non shared gene clusters, involving process that activates or increases the frequency, rate or extent of cellular DNA-templated transcription. The existence of non-shared gene clusters indicated that the biological process of this strain might have unique advantages over other related strains.
The specific primers (F-5’-GTCAGTCGGCGGCTTATTC-3’ and R-5’- ATATCGTCTCCTTGGCAATGTC-3’) of gene 1983 of chromosome 1 in the WY7 non shared gene clusters were designed for PCR detection of Brucella sp. WY7 and its relatives, Brucella anthropi ATCC® 49188TM (Figure S1). The results show that only WY7 has the correct band, which means that Brucella anthropi ATCC® 49188TM does not contain this gene, and Brucella sp. WY7 we found in Antarctic krill is not completely the same as its relatives.
(Please see the attachment.)
Fig. 6 The Venn diagram and the bar graph depict the comparative genomics among the genomes of Brucella anthropi ATCC® 49188TM (green), Brucella anthropi PBO (blue), Brucella anthropi NCTC® 12168TM (pink), Brucella melitensis bv. 1 str. 16M (yellow), and Brucella sp. WY7 (orange), showing shared and unshared orthologous genes clusters.

Reviewer 2 Report
Comments to authors:
Title and abstract:
1- The title can be improved, for example:"Genomic Characteristics and Functional Analysis of Ochrobactrum sp. strain WY7 Isolated from Antarctic Krill"
2- Please indicate in the abstract the origin of the strain Ochrobactrum anthropi ATCC 49188
3- ATCC strains should be written as follows: ATCC® 49188TM
4- Please avoid using “and so on”. Write down the most important ones only.
5- This sentence makes no sense “ suggesting that the strain WY7 has an aromatic compound as a potential metabolite.”. Please rewrite. May be: can use aromatic compounds in his metabolism ?.
6- Please, rewrite this sentence:
“Our work would help to understand the genomic characteristics and metabolic potential of Antarctic krill isolates”
For example: Our work would help to understand the genomic characteristics and metabolic potential of bacterial strains isolated from Antarctic krill
Introduction, MM, Discussion.
Line 49: Please, rewrite this sentence avoiding “and so on”. For example; “and affiliation to twelve genera, INCLUDING Bacillus, Delftia, Psychrobacter, Acinetobacter, and Pseudoalteromonas.
Line 62: Delete space between animals.__ According
Line 62: Please, change “According to studies” by “according to some published studies”
Lines 85-88. Author say: “The inoculated medium was cultured 85 at 10℃ until colonies were visible. The colonies were randomly isolated from medium, 86 were collected and expanded for about three times under the same conditions. The strains 87 were stored in a liquid medium containing 20% glycerol in refrigerator at –80℃ for sub-88 sequent use”.
How many colonies grew? Were they different? Please indicate why only a single colony/strain was ultimately selected for the study. Is this strain representative of all colonies isolated from the samples? Please discuss this in the discussion section. This strain may not be representative of the krill microbiota and other strains may be more important
Linea 90: What does "by the inoculation ring" mean? May be inoculation loop ?
Line 93: Please, indicate the concentration or % of the compound.
Lien 97: Please, change: The total DNA of genome by Genomic DNA
Line 101: Please, change to compared by for comparison
Lines 102-103. Please, rewrite. i.e. ClustalW program was used to analyze phylogeny [18].
Line 115. Ochrobactrum in italics.
Lines 148 and so on ATCC strains should be named as: ATCC® 49188TM
Line 178. Please, indicate the origin of strain Brucella lupini LUP21
Table 2. Please, indicate source or reference for these strains.
Table 3. Please, indicate source or reference for these strains.
Figure 2B and table 4. Please indicate what correlation exists between the data in parentheses of the subsystems in Figure 2B and Table 4, as it appears that the numbers are not equal. For example, 228 carbohydrate subsystems appear in the figure, but these numbers do not appear in the text and in table 4 (175 [118+57]). Also, other numbers of subsystems are unequal.
Figure 3: The letters in Figure 3 are not read correctly. Please increase the size or convert this figure to additional material with higher resolution. Please, increase the size or resolution of all figures in the manuscript.
Table 5. Please indicate your interest in knowing the "Organismal Systems" data of a bacterium.
English does not have many spelling mistakes but could be improved for easier reading.
Author Response
Response to Reviewer 2 Comments
Title and abstract:
Comment 1- The title can be improved, for example: "Genomic Characteristics and Functional Analysis of Ochrobactrum sp. strain WY7 Isolated from Antarctic Krill"
Response 1- Thank you very much for your suggestion. According to Reviewer-1 and your suggestions, the title was changed to “Genomic characteristics and functional analysis of Brucella sp. strain WY7 isolated from Antarctic krill”.
Comment 2- Please indicate in the abstract the origin of the strain Ochrobactrum anthropi ATCC 49188
Response 2- Thanks for your suggestion. The origin of the strain was added in the abstract. The sentence “The DNA-DNA hybridization value and average nucleotide identity value of strain WY7 and Ochrobactrum anthropi ATCC 49188 were 94.8% and 99.07%, respectively” was changed to “The DNA-DNA hybridization value and average nucleotide identity value of strain WY7 and Brucella anthropi ATCC® 49188TM, a type strain isolated from human clinical specimens were 94.8% and 99.07%, respectively” (lines 23-25).
Comment 3- ATCC strains should be written as follows: ATCC® 49188TM
Response 3- Thank you very much for your reminder. The “ATCC 49188” was changed to “ATCC® 49188TM” (line 24).
Comment 4- Please avoid using “and so on”. Write down the most important ones only.
Response 4- Thanks for your kind suggestion. The sentence “AntiSMASH analysis results showed that strain WY7 might produce many secondary metabolites, such as terpene, siderophore, ectoine, and so on.” was changed to “AntiSMASH analysis results showed that strain WY7 might produce many secondary metabolites, such as terpene, siderophore and ectoine.“ in the abstract (lines 27-29).
Comment 5- This sentence makes no sense “suggesting that the strain WY7 has an aromatic compound as a potential metabolite.”. Please rewrite. May be: can use aromatic compounds in his metabolism?
Response 5- Thanks for your suggestion. The sentence “suggesting that the strain WY7 has an aromatic compound as a potential metabolite.” was changed to “suggesting that the strain WY7 can use aromatic compounds in its metabolism.” (lines 30-31).
Comment 6- Please, rewrite this sentence:
“Our work would help to understand the genomic characteristics and metabolic potential of Antarctic krill isolates”
For example: Our work would help to understand the genomic characteristics and metabolic potential of bacterial strains isolated from Antarctic krill
Response 6- Thank you very much for your kind suggestion. The sentence “Our work would help to understand the genomic characteristics and metabolic potential of Antarctic krill isolates” was changed to “Our work would help to understand the genomic characteristics and metabolic potential of bacterial strains isolated from Antarctic krill” (lines 31-32).
Introduction, MM, Discussion.
Comment 7- Line 49: Please, rewrite this sentence avoiding “and so on”. For example; “and affiliation to twelve genera, INCLUDING Bacillus, Delftia, Psychrobacter, Acinetobacter, and Pseudoalteromonas.
Response 7- Thank you. The sentence “and affiliation to twelve genera, namely, Bacillus, Delftia, Psychrobacter, Acinetobacter, Pseudoalteromonas, and so on” was changed to “and affiliation to twelve genera, including Bacillus, Delftia, Psychrobacter, Acinetobacter, and Pseudoalteromonas” (lines 48-49).
Comment 8- Line 62: Delete space between animals.__ According
Line 62: Please, change “According to studies” by “according to some published studies”
Response 8- Thank you very much for your kind suggestions. We deleted the space between “animals.__ According”, and changed “According to studies” by “According to some published studies” (line 61).
Comment 9- Lines 85-88. Author say: “The inoculated medium was cultured at 10℃ until colonies were visible. The colonies were randomly isolated from medium, were collected and expanded for about three times under the same conditions. The strains were stored in a liquid medium containing 20% glycerol in refrigerator at –80℃ for sub-88 sequent use”.
How many colonies grew? Were they different? Please indicate why only a single colony/strain was ultimately selected for the study. Is this strain representative of all colonies isolated from the samples? Please discuss this in the discussion section. This strain may not be representative of the krill microbiota and other strains may be more important.
Response 9- Thanks for your questions. A total of 86 colonies were obtained from the Antarctic krill, belonging to 8 genera, including Delftia, Psychrobacter, Acinetobacter, Pseudoalteromonas, Jeotgalicoccus, Planococcus, Rhdococcus, and Brucella (Wang et al., 2022). Among them, 2 strains belonged to the Brucella genus, with a 100% sequence similarity in their 16S DNA sequences, confirming them as the same bacterial species. Upon reviewing relevant research on Antarctic isolated bacteria, we found a relative lack of studies concerning the Brucella genus. Therefore, we conducted comprehensive genomic research on the strain WY7 of this genus, aiming to fill the research gap related to polar bacteria. However, we acknowledge the importance of discussing this in more depth within the discussion section. It's crucial to address the potential limitations of using a single strain and its representativeness of the entire krill microbiota. We will expand on this aspect, highlighting the possibility of other strains playing significant roles within the krill microbiota. Based on your suggestion, we supplemented the sentences in the text as follows.
The sentence “The strains were stored in a liquid medium containing 20% glycerol in refrigerator at –80℃ for subsequent use.“ was added as “A total of 86 colonies were obtained, and the strains were stored in a liquid medium containing 20% glycerol in refrigerator at –80℃ for subsequent use.” (lines 87-88).
The sentence in discussion “In this study, we isolated and identified a strain Ochrobactrum sp. WY7, belonging to Alphaproteobacteria class, from Antarctic krill, and explored its genomic characteristics and potential functions.” was added as “In this study, we isolated and identified a strain Brucella sp. WY7, belonging to the genus not previously discovered in Antarctica, from Antarctic krill, and explored its genomic characteristics and potential functions.” (lines 330-333).
Comment 10- Line 90: What does "by the inoculation ring" mean? May be inoculation loop?
Response 10- Thanks for your question. The “inoculation ring” was changed to “inoculation loop” (line 90).
Comment 11- Line 93: Please, indicate the concentration or % of the compound.
Response 11- Thanks for your reminder. This sentence was supplemented as “negatively stained with 2% sodium phosphotungstate solution” (line 93).
Comment 12- Line 97: Please, change: The total DNA of genome by Genomic DNA
Response 12- Thanks for your suggestion. “The total DNA of genome” was changed to “Genomic DNA” (line 96).
Comment 13- Line 101: Please, change to compared by for comparison
Response 13- Thanks for your suggestion. “To compared” was changed to “for comparison” (line 100).
Comment 14- Lines 102-103. Please, rewrite. i.e. ClustalW program was used to analyze phylogeny [18].
Response 14- Thank you very much for your suggestion. The sentence “ClustalW program was used for sequencing against the closest type strains [18] to analyze phylogeny.” was changed to “ClustalW program in MEGA-X was used to analyze phylogeny [18-20].” (lines 101-102).
Comment 15- Line 115. Ochrobactrum in italics.
Response 15- Thank you for your reminder. The sentence “The genome information of the bacteria has been uploaded to NCBI, and accession numbers are CP049796-CP049798 for Ochrobactrum sp.” was changed to “The genome information of the bacteria has been uploaded to NCBI, and accession numbers are CP049796-CP049798 for Brucella sp.” (lines 110-111).
Comment 16- Lines 148 and so on ATCC strains should be named as: ATCC® 49188TM
Response 16- Thank you very much for your suggestion. The “ATCC 49188” were changed to “ATCC® 49188TM” (lines 144, 147, 173, 176, 297, 310, 311, 316, 340, 342).
Comment 17- Line 178. Please, indicate the origin of strain Brucella lupini LUP21
Table 2. Please, indicate source or reference for these strains.
Table 3. Please, indicate source or reference for these strains.
Response 17- Thank you very much for your suggestions. The origin of these strains were uniformly labeled in the Table 2 as follow (lines 179-180).
Table 2. The dDDH values are provided along with their confidence intervals (C.I.).
|
Subject strain |
query strain |
dDDH (d4, in%) |
C.I. (d4, in%) |
the origin of subject strain |
|
Brucella anthropi ATCC 49188 (JAAVLS010000001.1) |
WY7 |
94.8 |
[93.2-96.1] |
type strain from human clinical specimens, blood cultures |
|
Brucella anthropi NCTC 12168 (UGSA01000001.1) |
WY7 |
94.8 |
[93.2-96.1] |
type strain from human clinical specimens, blood cultures |
|
Brucella lupini LUP21 (GCA 002252535.1) |
WY7 |
82.5 |
[79.7-85.1] |
plant root nodule samples |
|
Brucella anthropi PBO (GCA 015326295.1) |
WY7 |
78.8 |
[75.9-81.5] |
plastics debris from sea coast, Qingdao, China |
|
Brucella melitensis 16M (GCA 000007125.1) |
WY7 |
25.0 |
[22.7-27.5] |
an infected goat |
Comment 18- Figure 2B and table 4. Please indicate what correlation exists between the data in parentheses of the subsystems in Figure 2B and Table 4, as it appears that the numbers are not equal. For example, 228 carbohydrate subsystems appear in the figure, but these numbers do not appear in the text and in table 4 (175 [118+57]). Also, other numbers of subsystems are unequal.
Response 18- Thank you very much for your serious reminder. We unified the data as shown in Figure 2B and deleted former Table 4. Also, we revised the expression in the text as follows (lines 183-202).
3.3. RAST genome quick annotation
The functional distribution of strain WY7 was preliminarily obtained by genome annotation with RAST. According to the SEED system of RAST, genes were assigned to subsystems, which could be divided into 27 categories. The genome annotation information showed that RAST divided them into 339 subsystems, only 28% (1256 genes) were annotated in the subsystem, and the other 72% of the genome were not assigned to the RAST subsystem. Among them, Amino Acids and Derivatives (359 genes), Carbohydrates (228 genes), Protein Metabolism (212 genes), Cofactors, Vitamins, Prosthetic Groups, Pigments (153 genes), Respiration (108 genes) and other functions accounted for the majority (Figure 2B). Above results showed that strain WY7 had rich amino acid and its derivatives, carbohydrates and protein metabolism.
Notably, the gene related to Cell Division and Cell Cycle was no found in strain WY7, but a gene related to Dormancy and Sporulation was found. The genome also lacked genes related to Photosynthesis, suggesting that the bacterium was not a photosynthetic bacterium.
(Please see the attachment.)
Figure 2. A Genome circle map of strain Brucella sp. WY7: GC content (oxblood red), GC skew curves (+/−, dark green/purple), open reading frames (ORFs, light blue), coding sequences (CDSs, dark blue), rRNAs (light green), tRNAs (vermilion), and repeat (pink). B RAST subsystems category distribution of annotated genes of the strain Brucella sp. WY7.
Comment 19- Figure 3: The letters in Figure 3 are not read correctly. Please increase the size or convert this figure to additional material with higher resolution. Please, increase the size or resolution of all figures in the manuscript.
Response 19- Thank you for your kind suggestions. We rewrote the unclear letters in Figure 3 as follow and increased the resolution of all figures.
(Please see the attachment.)
Figure 3. A KEGG annotation statistics chart of Brucella sp. WY7. Figure shows chromosome 1, chromosome 2 and plasmid of Brucella sp. WY7 respectively. The horizontal axis is the number of genes, the vertical axis represents the name of the Level 2 pathway. B Aromatic hydrocarbon degradation pathways in WY7.
Comment 20- Table 5. Please indicate your interest in knowing the "Organismal Systems" data of a bacterium.
Response 20- Thank you for your suggestion. Table 5 was changed to Table 4. We included functional annotation of genomic KEGG database in Table 4 including the classification "Organismal Systems". The "Organismal Systems" data provides valuable insights into how individual components collaborate to ensure the bacterium's survival, growth, and response to its environment. By exploring this data, we can uncover the underlying mechanisms that govern various cellular processes, metabolic pathways, and regulatory networks. This holistic perspective is crucial for deciphering the bacterium's behavior and its impact within larger ecosystems. We look forward to exploring this aspect further. Considering your suggestion, we added a description about Organizational Systems in the Results section.
The sentence in Results “The results showed that strain WY7 had strong carbohydrate metabolism and membrane transport capacity, thus providing great potential for the degradation, transformation, and utilization of complex carbohydrates.” was added as “A total of 40 genes in WY7 are associated with Organismal Systems, mainly involving the Endocrine system, Aging, Environmental adaptation, Neurous system, Circulatory system, and Digestive system. The results showed that strain WY7 had strong carbohydrate metabolism and membrane transport capacity, and might also have an impact on the Organizational Systems of its host, thus providing great potential for environmental adaptation and the degradation, transformation, and utilization of complex carbohydrates.” (lines 216-222).
